# Neuroprotective Potential of Raloxifene via G-Protein-Coupled Estrogen Receptors in Aβ-Oligomer-Induced Neuronal Injury

**DOI:** 10.3390/biomedicines11082135

**Published:** 2023-07-28

**Authors:** Tetsuhito Nohara, Mayumi Tsuji, Tatsunori Oguchi, Yutaro Momma, Hideaki Ohashi, Miki Nagata, Naohito Ito, Ken Yamamoto, Hidetomo Murakami, Yuji Kiuchi

**Affiliations:** 1Division of Medical Pharmacology, Department of Pharmacology, School of Medicine, Showa University, Tokyo 142-8555, Japan; t-nohara2@med.showa-u.ac.jp (T.N.); t.oguchi@med.showa-u.ac.jp (T.O.); m-yutaro@med.showa-u.ac.jp (Y.M.); hitto226@med.showa-u.ac.jp (N.I.); ken_yamamoto@med.showa-u.ac.jp (K.Y.); ykiuchi@med.showa-u.ac.jp (Y.K.); 2Division of Neurology, Department of Internal Medicine, School of Medicine, Showa University, Tokyo 142-8555, Japan; mimixart@yahoo.co.jp (H.O.); hidneu@med.showa-u.ac.jp (H.M.); 3Pharmacological Research Center, Showa University, Tokyo 142-8555, Japan; 4Department of Hospital Pharmaceutics, School of Pharmacy, Showa University, Tokyo 142-8555, Japan; miki042@cmed.showa-u.ac.jp

**Keywords:** Aβ oligomers, raloxifene, Alzheimer’s disease, GPER, neurotoxicity, oxidative stress

## Abstract

Amyloid-β (Aβ) is one of the causes of Alzheimer’s disease (AD), damaging nerve membranes and inducing neurotoxicity. AD is more prevalent in female patients than in male patients, and women are more susceptible to developing AD due to the decline in estrogen levels around menopause. Raloxifene, a selective estrogen receptor modulator, exhibits protective effects by activating the transmembrane G-protein-coupled estrogen receptor (GPER). Additionally, raloxifene prevents mild cognitive impairment and restores cognition. However, the influence of raloxifene via GPER on highly toxic Aβ-oligomers (Aβo)-induced neurotoxicity remains uncertain. In this study, we investigated the GPER-mediated neuroprotective effects of raloxifene against the neurotoxicity caused by Aβo-induced cytotoxicity. The impact of raloxifene on Aβo-induced cell damage was evaluated using measures such as cell viability, production of reactive oxygen species (ROS) and mitochondrial ROS, peroxidation of cell-membrane phospholipids, and changes in intracellular calcium ion concentration ([Ca^2+^]_i_) levels. Raloxifene hindered Aβo-induced oxidative stress and reduced excessive [Ca^2+^]_i_, resulting in improved cell viability. Furthermore, these effects of raloxifene were inhibited with pretreatment with a GPER antagonist. Our findings suggest that raloxifene safeguards against Aβo-induced neurotoxicity by modifying oxidative parameters and maintaining [Ca^2+^]_i_ homeostasis. Raloxifene may prove effective in preventing and inhibiting the progression of AD.

## 1. Introduction

The prevalence of dementia is expected to rise as life expectancy increases and the elderly population grows due to advancements in medical technology. It is estimated to reach 175.6 million people worldwide by 2050, up from 130.8 million [1], making it a major medical and social concern. Alzheimer’s disease (AD) is the most common form of dementia, affecting over 4% of adults aged 60 and above [2].

AD is a disorder characterized by initial short-term memory loss and disorientation, progressing to psychiatric symptoms such as wandering, irritability, and higher brain dysfunction, eventually leading to bedridden patients within approximately 10 years [3]. The cause of AD remains unknown, but it is pathologically characterized by the formation of senile plaques, which are aggregates of amyloid-β (Aβ), and neurofibrillary tangles composed of tau protein [4].

Aβ is a protein consisting of approximately 40 amino acids, with Aβ_1–42_ being highly abundant in AD pathology and considered the primary cause of the disease. Previous studies have shown that Aβ_1–42_ is produced from amyloid precursor protein (APP) in its monomeric form and easily aggregates to form highly toxic Aβ_1–42_ oligomers [5]. Lecanemab, a monoclonal antibody targeting soluble Aβ protofibrils (high-molecular-weight Aβ oligomers (Aβo)), has been found to inhibit the progression of Clinical Dementia Rating Sum of Boxes (CDR-SB) by approximately 27% in patients with mild cognitive impairment and early AD (Clarity AD ClinicalTrials.gov, number NCT03887455). It was granted urgent approval by the US Food and Drug Administration (FDA) in 2023 as a treatment for AD [6].

Epidemiologically, AD is approximately twice as common in female patients as in male patients, and the decrease in endogenous estrogen production after menopause is thought to be involved in the development of AD [7]. Clinical trials have reported a decreased risk of AD in women who received estrogen replacement therapy [8]. Estrogen is believed to exert several neuroprotective effects on the aging brain, including inhibiting Aβ formation, enhancing cholinergic activity, and reducing cellular damage caused by oxidative stress [9].

Additionally, patients with AD have a higher incidence of fractures [10] and decreased bone mineral density (BMD) [11]. It has also been reported that Aβ enhances RANKL-induced osteoclast activation in vivo via osteoclast differentiation and calcium signaling [12], suggesting a shared pathogenic mechanism between AD and osteoporosis.

Osteoporosis was first reported by Albright et al. in 1941 [13], and conjugated estrogens (Premarin^®^), a treatment for menopausal symptoms, were approved by the FDA in 1988 for the prevention of osteoporosis. However, estrogen replacement therapy has been shown to increase the risk of estrogen-dependent tumors such as uterine cancer [14]. Recently, raloxifene, a selective estrogen receptor modulator (SERM) that does not increase the risk of estrogen-dependent tumors, has been widely used for osteoporosis in early postmenopausal women. Estrogen and raloxifene have an agonist effect on bone tissue and prevent the progression to osteoporosis by regulating osteoclast differentiation and activation, which is opposite to the effect of Aβ on osteoclasts. Therefore, it is assumed that they are effective in the development of AD based on epidemiological and pharmacological evidence. It has been inferred that raloxifene influences the onset of AD, as it has been reported to inhibit Aβ aggregation [15]. However, there are scattered reports from retrospective studies regarding the oral administration of raloxifene and its impact on cognitive function. Some studies suggest that it preserves cognitive function [16], while others indicate that it has no effect [17]. Recently, large-scale clinical trials were conducted in patients with mild cognitive impairment and AD, and the results demonstrated that raloxifene was not effective in preventing the onset of AD [18]. However, it is known that Aβ aggregation has already accumulated in brain tissue during the stage of cognitive decline [19], and the effects of raloxifene have not been fully understood. Therefore, this study aimed to compare the protective effect of raloxifene against Aβo-induced neuronal injury with that of estradiol in vitro and investigate whether the protective effect is mediated by the transmembrane G-protein-coupled estrogen receptor (GPER).

## 2. Materials and Methods

### 2.1. Reagents and Cells

Aβ_1–42_ and Aβ_1–40_ were purchased from Peptide Institute Inc. (Osaka, Japan), and SH-SY5Y cells (human neuroblastoma; EC94030304) were obtained from the European Collection of Authenticated Cell Cultures (London, UK). Dulbecco’s Modified Eagle Medium (DMEM) Ham’s/F-12 medium and all-trans retinoic acid (ATRA) were obtained from Fuji Film Wako Pure Chemical Co., Ltd. (Osaka, Japan). Penicillin G sodium, streptomycin sulfate, amphotericin B, and fetal bovine serum (FBS) were purchased from Thermo Fisher Scientific K.K. (Waltham, MA, USA). 17β-estradiol and G-15 were procured from Cayman Chemical Company (Ann Arbor, MI, USA), and raloxifene hydrochloride was obtained from Tokyo Chemical Industry Co., Ltd. (Tokyo, Japan). The other chemicals used in this experiment were commercially available and of the purest grade. Their structures are shown below (Figure 1). Special-grade products were used for other reagents.

### 2.2. Aggregation Kinetics of Aβ_1–42_ and Aβ_1–40_

The aggregation kinetics of Aβ_1–42_ and Aβ_1–40_ were measured using a SensoLyte Thioflavin T β-Amyloid (1–42) Aggregation Kit and a SensoLyte Thioflavin T β-Amyloid (1–40) Aggregation Kit (AnaSpec, Inc., Fremont, CA, USA). Thioflavin T (ThT) emits little fluorescence when present free in solution, but the dye emits very strong fluorescence when bound to beta-sheet-rich amyloid structures. To measure the aggregation kinetics of Aβ_1–42_ or Aβ_1–40_ in a 96-well black microplate, 10 μL ThT (2 mM), and 5 μL of raloxifene and estradiol dissolved in dimethyl sulfoxide (DMSO) were added to each well, and 85 μL of Aβ solution was mixed. The final concentration of Aβ_1–42_ peptide was 25 µM; those of raloxifene were 1, 5, 10, 20, and 50 µM; and those of estradiol were 1, 5, 10, and 20 µM. ThT fluorescence intensity was monitored at 37 °C for 2 h (Aβ_1–42_) or 6 h (Aβ_1–40_) at 15 min intervals with an excitation wavelength of 440 nm and an emission wavelength of 484 nm using a SpectraMax i3 microplate reader (Molecular Devices, LLC., Sunnyvale, CA, USA). The experiments were performed in triplicate.

### 2.3. Separation and Collection of Aβ Molecular Species with HPLC

Aβo, which is the most toxic form of Aβ, was isolated and collected. Aβ_1–42_ peptides were dissolved in 10 mM NaOH and sonicated for 2 min; then, a 10-fold concentration of phosphate buffer was added to bring the concentration to 500 µM. The Aβ_1–42_ solution was passed through a Millex^®^-LG filter (0.20 μm; Millipore Ireland BV, Dublin, Ireland) to remove high-molecular-weight substances and incubated at 37 °C for 1 h. After incubation, the Aβ_1–42_ solutions were centrifuged at 15,000 rpm for 5 min. The supernatant was collected, and the protein concentration was measured using Bio-Rad Protein Assay Dye Reagent Concentrate (Bio-Rad Laboratories, Inc., Hercules, CA, USA). The supernatant was diluted to 50 µM with phosphate buffer, and size exclusion chromatography was used to confirm the presence of Aβo. A Superdex 75 increase 10/300 GL column was employed, and the supernatant was fractionated at a flow rate of 0.8 mL/min. The peak of Aβo appeared at 10 min, confirming that Aβo occupied the majority of the Aβ solution (Figure 2). Previous reports have identified this peak as Aβo [20,21].

### 2.4. SH-SY5Y Cell Culture and the Reagent Treatment Method

SH-SY5Y cells were cultured in DMEM/Ham’s F-12 medium containing 10% FBS. Differentiation was then induced by culturing for 7 days in medium containing ATRA at a final concentration of 10 μM. Raloxifene and estradiol were dissolved in DMSO to achieve a final DMSO concentration of 0.1%. In the subsequent experiments, cells treated with 0.1% DMSO alone were used as the control.

### 2.5. Effect of Raloxifene and Estradiol on Aβo-Induced Cytotoxicity

#### 2.5.1. Detection of Viability via the MTT Assay

The viability of treated SH-SY5Y cells was evaluated using MTT (3-(4,5-dimethylthiazol-2-yl)-2,5-diphenyltetrazolium bromide) Cell Proliferation kit I (Nacalai Tesque Inc., Kyoto, Japan). Differentiated SH-SY5Y cells were seeded at a concentration of 1.0 × 10^6^ cells/mL onto 96-well collagen-coated plates and incubated at 37 °C for 24 h. First, SH-SY5Y cells were exposed to 0.5 to 10 μM Aβo for 3 h to determine the optimal concentration of Aβo for cytotoxicity.

Next, to examine the protective effect of raloxifene and estradiol against Aβo-induced cytotoxicity, SH-SY5Y cells were treated with Aβo + raloxifene (0.5, 1, and 5 μM) or Aβo + estradiol (0.5, 1, and 5 μM) for 3 h. The effects of GPER and estrogen receptor (ER) on the protective effects of raloxifene and estradiol were confirmed using G-15 (GPER antagonist) and fulvestrant (ER antagonist), respectively. SH-SY5Y cells pretreated with G-15 for 30 min and fulvestrant for 30 min were then treated with Aβo, Aβo + raloxifene, or Aβo + estradiol for 3 h. After incubation, the MTT assay was performed, and the absorbance was measured at 570 nm using SpectraMax i3 (Molecular Devices). The viability of SH-SY5Y cells incubated with 10% FBS-containing medium was showed as 100% and the viability of cells exposed to 1% saponin was expressed as 0%.

#### 2.5.2. Detection of Cell Viability and Cytotoxicity Using Calcein-AM/Ethidium Homodimer-1 (Live/Dead) Cell Assay

Cell viability was measured by simultaneously assessing live and dead cells using calcein-AM and ethidium homodimer-1 (Thermo Fisher Scientific K.K.). Live cells were observed with calcein-AM and dead cells with ethidium homodimer-1. Differentiated SH-SY5Y cells were seeded at a concentration of 1.0 × 10^6^ cells/mL onto 96-well collagen-coated plates and incubated at 37 °C for 24 h. Cells were then exposed to Aβo or treated with Aβo + 5 μM raloxifene or Aβo + 5 μM estradiol for 3 h. The treated cells were stained with 2 μM calcein-AM and 10 μM ethidium homodimer-1. Calcein-AM is hydrolyzed by ubiquitous intracellular esterases, resulting in green fluorescence that is proportional to the number of viable cells. Ethidium homodimer-1 penetrates only into cells with damaged membranes, binds to nucleic acids, and exhibits red fluorescence in proportion to the number of dead cells. Red fluorescence intensity was measured at excitation (Ex) of 495 nm and emission (Em) of 645 nm, while the green fluorescence intensity was measured at Ex of 495 nm and Em of 530 nm using SpectraMax i3 (Molecular Devices). Additionally, the cytotoxicity of individual cells was assessed by observing with a fluorescence microscope (BZX800; Keyence Co., Osaka, Japan). Red fluorescence intensity is also represented as gray-scale images.

### 2.6. Detection of Oxidative Stress

#### 2.6.1. Reactive Oxygen Species (ROS)

Oxidative stress was measured using 5-(and-6)-chloromethyl-2’,7’-dichlorodihydrofluorescein diacetate (CM-H2DCFDA; Thermo Fisher Scientific Inc.). CM-H2DCFDA is taken up intracellularly and converted to 2′,7′-dichlorofluorescein (DCF) by esterase. DCF is then oxidized by intracellular ROS and exhibits green fluorescence. Differentiated SH-SY5Y cells were seeded in 96-well collagen-coated plates at a concentration of 1.0 × 10^6^ cells/mL and incubated at 37 °C for 24 h. After incubation, the cells were pretreated with 5 μM G-15 for 10 min and then treated with Aβo, Aβo + 5 μM raloxifene, or Aβo + 5 μM estradiol for 3 h. ROS production was measured using the SpectraMax i3 microplate reader at Ex of 480 nm and Em of 530 nm. Following fluorescence measurements, the cellular protein content in the plate was determined using Bio-Rad Protein Assay Dye Reagent Concentrate (Bio-Rad Laboratories, Inc., Hercules, CA, USA), and the results were represented as fluorescence intensity per unit protein. Furthermore, the oxidative stress state was observed with a fluorescence microscope (BZX800; Keyence Co.).

#### 2.6.2. Mitochondrial ROS Production

Mitochondrial ROS contribute significantly to intracellular ROS. Intracellular mitochondrial ROS were detected using a Mitochondrial ROS detection kit (701600; Cayman Chemical Company, Ann Arbor, MI, USA). Differentiated SH-SY5Y cells were seeded in 96-well collagen-coated plates at a density of 1.0 × 10^6^ cells/mL and incubated at 37 °C for 24 h. After incubation, the cells were pretreated with 5 µM G-15 for 30 min and then treated with Aβo, Aβo + 5 µM raloxifene, or Aβo + estradiol for 3 h and 24 h. Following fluorescence measurements, the cellular protein content in the plate was determined using Bio-Rad Protein Assay Dye Reagent Concentrate (Bio-Rad Laboratories, Inc.), and the results were represented as fluorescence intensity per unit protein. Fluorescence intensity was measured at Ex of 500 nm and Em of 580 nm using Spectra Max i3 (Molecular Devices).

#### 2.6.3. Measurement of Phospholipid Peroxidation in Cell Membranes

Lipid peroxidation in cell membranes was measured using diphenyl-1-pyrenyl phosphine (DPPP) (Dojin Chemical Laboratory, Kumamoto, Japan), a compound with high selectivity for hydroperoxides. DPPP does not fluoresce on its own but reacts quantitatively with hydroperoxides, emitting strong fluorescence. Differentiated SH-SY5Y cells were seeded at a density of 1.0 × 10^6^ cells/mL in 96-well collagen-coated plates and incubated at 37 °C for 24 h. After incubation, the cells were pretreated with 5 µM G-15 for 30 min and then treated with Aβo, Aβo + 5 µM raloxifene, or Aβo + estradiol for 30 min and 3 h. Subsequently, 50 µM DPPP was added to the cells and incubated at 37 °C for 10 min. Fluorescence measurements were made using Spectra Max i3 (Molecular Devices) at Ex of 352 nm and Em of 380 nm. Following fluorescence measurements, the cellular protein content on the plate was determined using Bio-Rad Protein Assay Dye Reagent Concentrate (Bio-Rad Laboratories, Inc.), and the results were represented as fluorescence intensity per unit protein. Additionally, the phospholipid peroxidation state was observed using a fluorescence microscope (BZX800; Keyence Co.). Fluorescence intensity values are shown as gray-scale images.

### 2.7. Measurement of [Ca^2+^]_i_ Changes

Changes in intracellular calcium ([Ca^2+^]_i_) levels in SH-SY5Y cells were measured using FLIPR Calcium 5 Assay Kit (Molecular Devices). Differentiated SH-SY5Y cells were loaded with FLIPR reagent containing 20 mM HEPES and Hank’s Balanced Salt Solution (pH 7.4) for 60 min at 37 °C. After 20 s of [Ca^2+^]_i_ measurement, Aβo, 5 µM raloxifene, and 5 µM estradiol were added, and changes in [Ca^2+^]_i_ were measured.

Next, the effects of raloxifene and estradiol on [Ca^2+^]_i_ changes following Aβo exposure were examined by pretreating the cells with raloxifene and estradiol for 10 min. Similarly, to investigate whether the effects of Aβo on [Ca^2+^]_i_ are mediated by Ca channels or NMDA receptors, the cells were pretreated with 10 μM nicardipine (an L-type voltage-gated Ca channel blocker), or 10 μM MK801 (an NMDA receptor blocker) for 10 min, and changes in [Ca^2+^]_i_ following Aβo exposure were measured.

In addition, 5 μM G-15, a GPER blocker, was pretreated 10 min before the addition of raloxifene and estradiol. The changes in [Ca^2+^]_i_ were measured using fluorescence at Ex of 485 nm and Em of 525 nm every 5 s for 300 s at 37 °C using FlexStation 3 (Molecular Devices). The results were expressed as 100% of the starting fluorescence intensity.

### 2.8. Statistical Analysis

Each measurement was performed three times, and the experimental results are presented as means ± standard errors of the mean (SEMs). The effects of raloxifene and estradiol were compared with Aβo using Tukey’s or Dunnett’s post hoc test after analysis of variance (ANOVA). Values with *p* < 0.05 were considered statistically significant.

## 3. Results

### 3.1. Effects of Raloxifene and Estradiol on Aβ_1–42_ and Aβ_1–40_ Aggregation

A ThT fluorescence assay was used to compare the effects of raloxifene and estradiol on the aggregation kinetics of Aβ_1–42_ and Aβ_1–40_ peptides.

The aggregation kinetics of the Aβ_1–42_ peptide was monitored for 120 min. Since Aβ readily aggregates to form Aβo, aggregation of Aβ_1–42_ with 0.1% DMSO alone increased exponentially without delay, reaching fluorescence intensity after 120 min that was approximately four times greater than that observed at the beginning.

However, in the presence of 20 μM raloxifene, the fluorescence intensity decreased in a concentration-dependent manner after 30 min of measurement (*p* = 0.0464 vs. Aβ_1–42_ + 0.1% DMSO). After 120 min, 20 μM raloxifene caused a 38% inhibition (*p* = 0.0014 vs. Aβ_1–42_ + 0.1% DMSO) (Figure 3A). On the other hand, the presence of estradiol did not show a significant effect on Aβ_1–42_ aggregation (Figure 3B). In experiments without the Aβ_1–42_ peptide, neither raloxifene nor estradiol treatment showed an increase in fluorescence intensity with time.

The aggregation rate of the Aβ_1–40_ peptide was monitored for 360 min since it aggregates more slowly than Aβ_1–42_. Like Aβ_1–42_, aggregation of Aβ_1–40_ with 0.1% DMSO increased exponentially without delay. Raloxifene significantly suppressed the increase in fluorescence intensity compared with Aβ_1–40_ + 0.1% DMSO at concentrations of 10 μM (*p* = 0.0490 vs. Aβ_1–40_ + 0.1% DMSO) and 20 μM (*p* = 0.0063 vs. Aβ_1–40_ + 0.1% DMSO) after 60 min. After 360 min, 20 μM raloxifene caused approximately 40% suppression (*p* = 0.0001 vs. Aβ_1–40_ + 0.1% DMSO) (Figure 3C). However, estradiol did not show a significant effect on Aβ_1–40_ aggregation (Figure 3D).

### 3.2. Effects of Raloxifene and Estradiol on Aβo-Induced Cytotoxicity

#### 3.2.1. Detection of Cell Viability via the MTT Assay

Raloxifene and estradiol were dissolved in DMSO, and 0.1% DMSO-treated cells were used as controls. Three hours of Aβo exposure in SH-SY5Y cells significantly decreased cell viability at all concentrations from 0.5 µM to 10 µM (*n* = 8, *p* < 0.0001, Dunnett’s) (Figure 4A). Based on this result, we decided to use 5 µM Aβo exposure, which exhibited approximately 50% viability compared with the control, for subsequent experiments.

The viability of cells exposed to 5 µM Aβo was significantly reduced by approximately 40% compared with the control (*n* = 10, Tukey’s, *p* < 0.0001). Raloxifene treatment at 1 µM (*p* = 0.0052) and 5 µM (*p* < 0.0001 vs. 5 µM Aβo) showed a significant increase in viability, while cells treated with 1 µM estradiol showed no significant effect, and 5 µM estradiol treatment showed a significant increase in cell viability (*p* = 0.049 vs. 5 µM Aβo) (Figure 4B).

Next, to confirm whether the protective effect of raloxifene against Aβo-induced cytotoxicity is mediated by GPER, the same experiments were performed with 5 µM G-15 pretreatment for 30 min, followed by 5 µM raloxifene, 5 µM estradiol, and 5 µM Aβo. G-15 pretreatment resulted in a decrease in viability compared with the G-15-untreated group for both raloxifene and estradiol (*p* < 0.0001 vs. G-15-untreated for each cell). No effect on viability was observed with G-15 pretreatment for 5 µM Aβo exposure (Figure 4C).

To determine whether the protective effect of raloxifene against Aβo-induced cytotoxicity is mediated by the ER, the same experiments were performed with 5 µM fulvestrant pretreatment for 30 min, followed by 5 µM raloxifene, 5 µM estradiol, and 5 µM Aβo treatment. Fulvestrant pretreatment did not significantly reduce survival compared with the untreated group for both raloxifene and estradiol (Figure 4D). These experiments suggest that upon exposure to Aβo for 3 h, the protective effect of raloxifene and estradiol against Aβo-induced cytotoxicity is not mediated by ER stimulation but rather by GPER stimulation.

#### 3.2.2. Detection of Cell Viability and Cytotoxicity Using the Calcein-AM and Ethidium Homodimer-1 (Live/Dead) Cell Assay

Figure 5 shows the results of the calcein-AM/ethidium homodimer-1 staining of SH-SY5Y cells treated with 5 µM Aβo for 3 h in the presence of raloxifene and estradiol. Cells exposed to Aβo exhibited a significant increase in cytotoxicity compared with the control (*n* = 10, *p* < 0.0001 vs. control, Tukey’s). Raloxifene treatment significantly suppressed Aβo-induced cytotoxicity (*p* = 0.0001 vs. 5 µM Aβo). Estradiol treatment showed a tendency to suppress Aβo-induced cytotoxicity (*p* = 0.0705 vs. 5 µM Aβo), although the difference was not significant (Figure 5A).

When the calcein-AM/ethidium homodimer-1 staining of SH-SY5Y cells was evaluated with fluorescence microscopy, increased red fluorescence indicative of dead cells was observed in cells exposed to Aβo (Figure 5G,K). In raloxifene-treated cells, a decrease in red fluorescence associated with Aβo exposure was observed (Figure 5H,I,L,M).

### 3.3. Effects of Raloxifene and Estradiol on Aβo-Induced Oxidative Stress

Aβo induces oxidative stress, and increased oxidative stress contributes to cell-membrane damage and cell death, suggesting that oxidative stress plays an important role in the pathogenesis of AD [22]. Next, the protective effects of raloxifene and estradiol against Aβo-induced oxidative stress were investigated.

#### 3.3.1. Measurement of ROS Production

SH-SY5Y cells exposed to 5 µM Aβo showed a significant increase in ROS production compared with the control (*n* = 10, *p* = 0.0086, Tukey’s), confirming the oxidative stress effect of Aβo. In addition, compared with Aβo, cells treated with raloxifene (*p* < 0.0001 vs. 5 μM Aβo) and estradiol (*p* = 0.0004 vs. 5 μM Aβo) showed a significant decrease in ROS production (Figure 6A).

To further confirm whether raloxifene’s inhibitory effect on oxidative stress induced by Aβo involves GPER, we conducted identical experiments with a 30 min pretreatment of 5 µM G-15, followed by 5 µM raloxifene, 5 µM estradiol, and 5 µM Aβo. Pretreatment with G-15 resulted in increased ROS production in both raloxifene + 5 µM Aβo (*p* = 0.0351) and estradiol + 5 µM Aβo (*p* = 0.0004 vs. G-15-untreated in each cell) compared with the G-15-untreated group. No effect on ROS production was observed with G-15 pretreatment for 5 µM Aβo exposure (Figure 6A).

Fluorescence microscopy images are depicted in Figure 6F–I, N–Q. Aβo exposure enhanced green dichlorofluorescein (DCF) fluorescence (Figure 6G), while raloxifene and estradiol treatments reduced green fluorescence (Figure 6H, I). When pretreated with G-15 and observed, both raloxifene + 5 µM Aβo and estradiol + 5 µM Aβo increased ROS generation compared with the G-15-untreated group (Figure 6P–Q).

#### 3.3.2. Mitochondrial ROS

SH-SY5Y cells exposed to 5 µM Aβo demonstrated a significant increase in mitochondrial ROS production at 3 h (*n* = 8, *p* = 0.0024, Tukey) and 24 h of exposure (*n* = 8, *p* = 0.0010, Tukey’s) compared with the control, confirming the oxidative stress effect of Aβo on mitochondria. At 3 h of exposure, a significant decrease in mitochondrial ROS levels was observed in cells treated with raloxifene (*p* < 0.0001 vs. 5 μM Aβo) and estradiol (*p* < 0.0001 vs. 5 μM Aβo) compared with those stimulated with Aβo alone (Figure 7A). At 24 h of exposure, cells treated with raloxifene (*p* = 0.0454 vs. 5 μM Aβo) exhibited a significant decrease in mitochondrial ROS compared with cells stimulated with Aβo alone, whereas cells treated with estradiol (*p* = 0.2059 vs. 5 μM Aβo) showed no significant decrease in mitochondrial ROS compared with cells stimulated with Aβo alone (Figure 7B).

To confirm whether raloxifene’s inhibitory effect on mitochondrial oxidative stress induced by Aβo is mediated by GPER, we performed the same experiment with a 30 min pretreatment with 5 µM G-15, followed by 5 µM raloxifene, 5 µM Aβo, and 5 µM estradiol. At 3 h of exposure, G-15 pretreatment caused an increase in mitochondrial ROS levels using raloxifene (*p* = 0.0001 vs. G-15-untreated raloxifene + Aβo-treated cells), while G-15 pretreatment caused no significant increase using estradiol (*p* = 0.8234 vs. G-15-untreated estradiol + Aβo-treated cells) compared with those in the G-15-untreated group. At 24 h of exposure, G-15 pretreatment caused an increase in mitochondrial ROS in the raloxifene (*p* < 0.0001 vs. G-15-untreated raloxifene + Aβo-treated cells) and estradiol (*p* = 0.0006 vs. G-15-untreated in each cell) groups compared with the G-15-untreated group. These results indicate that the inhibitory effects of raloxifene and estradiol on Aβo-induced oxidative stress are mediated by GPER. G-15 pretreatment tended to increase mitochondrial ROS levels in Aβo-only exposure (*p* = 0.0184 vs. G-15-untreated Aβo-only exposure) and Aβo (−) control (*p* = 0.0946 vs. G-15-untreated Aβo (−)control) groups.

#### 3.3.3. Detection of Cell-Membrane Phospholipid Peroxidation Capacity

Aβo is believed to directly bind to membrane lipids, damaging the phospholipid bilayer structure and entering cells [23]. In this study, we examined changes in plasma-membrane phospholipid peroxidation induced by Aβo exposure.

As shown in Figure 8A,B, there was a significant increase in plasma-membrane phospholipid peroxidation in cells exposed to 5 µM Aβo compared with the control at 30 min (*n* = 10, *p* = 0.0003, Tukey’s) and 3 h (*n* = 10, *p* = 0.0009, Tukey’s) of exposure. At 30 min of exposure, SH-SY5Y cells treated with raloxifene or estradiol exhibited significant inhibition of Aβo-induced plasma-membrane phospholipid peroxidation (Figure 8A). Moreover, at 3 h of exposure, SH-SY5Y cells treated with raloxifene exhibited significant inhibition of Aβo-induced plasma-membrane phospholipid peroxidation, and those treated with estradiol tended to inhibit Aβo-induced plasma-membrane phospholipid peroxidation (Figure 8B). To further confirm whether raloxifene’s inhibitory effect on increased plasma-membrane phospholipid peroxidation by Aβo involves GPER, we pretreated with 5 µM G-15 for 10 min, followed by 5 µM raloxifene + Aβo and 5 µM estradiol + Aβo for 30 min. G-15 pretreatment had no significant effect on plasma-membrane phospholipid peroxidation compared with the G-15-untreated group for both raloxifene and estradiol at 30 min and 3 h of exposure to Aβo. No effect on plasma-membrane phospholipid peroxidation was observed with G-15 pretreatment for 5 µM Aβo exposure.

Fluorescence microscopy images are shown in Figure 8G–J. DPPP reacted with hydroperoxides and emitted intense fluorescence, which was enhanced by Aβo exposure (Figure 8H) but reduced by raloxifene and estradiol treatments (Figure 8I,J). The fluorescence intensity values are also presented as gray-scale images.

### 3.4. Changes in [Ca^2+^]_i_ Levels following Treatment with Raloxifene and Estradiol

As depicted in Figure 9A, there was a rapid increase in [Ca^2+^]_i_ immediately after exposure to 5 µM Aβo, reaching a peak of 25.5% at 60 s and then maintaining a nearly constant increased level. When loaded with Ca^2+^-free buffer, no increase in [Ca^2+^]_i_ was observed after exposure to 5 µM Aβo. However, when pretreated with 10 µM Nicardipine and 10 µM MK801, [Ca^2+^]_i_ increased by only 10% immediately after the addition of 5 µM Aβo.

Upon exposure to raloxifene and estradiol, [Ca^2+^]_i_ temporarily increased by approximately 110% immediately after exposure but quickly decreased and maintained a constant [Ca^2+^]_i_ thereafter. When pretreated with G-15, there was no transient increase in [Ca^2+^]_i_ immediately after raloxifene and estradiol addition (Figure 9B).

Furthermore, Figure 9C,D presents the results of raloxifene and estradiol pretreatment on the increase in [Ca^2+^]_i_ caused by Aβo stimulation. The addition of 5 μM Aβo increased [Ca^2+^]_i_ by 131.2% at 65 s. Pretreatment with 5 μM raloxifene suppressed the increase in [Ca^2+^]_i_ caused by Aβo between 60 s (*p* = 0.0192 vs. 5 μM Aβo) and 300 s (*p* = 0.0036 vs. 5 μM Aβo), and [Ca^2+^]_i_ was further reduced in cells pretreated with 5 µM G-15 at 65 s (*p* = 0.0279 vs. 5 μM Aβo + Ral). Similarly, pretreatment with 5 µM estradiol decreased the [Ca^2+^]_i_ increase caused by Aβo between 60 s (*p* = 0.0005 vs. 5 μM Aβo) and 300 s (*p* = 0.0009 vs. 5 μM Aβo), and [Ca^2+^]_i_ showed a trend towards a further decrease in cells pretreated with 5 µM G-15 at 65 s (*p* = 0.749 vs. 5 μM Aβo + Est) (Figure 9C,D).

## 4. Discussion

In this experiment, we demonstrated that raloxifene provided protection against Aβo-induced neuronal cytotoxicity in SH-SY5Y cells. Aβ is produced as a monomer via sequential two-step cleavage of intracellularly produced APP by β-secretase1 and γ-secretase. The extracellularly secreted Aβ monomer then aggregates to form oligomers, which exhibit neuronal cytotoxicity [24]. For this study, we utilized high-molecular-weight Aβo, which is considered the most toxic form of Aβo [25].

Estrogen is a steroid hormone primarily produced in the granulosa cells of the ovary. As an estrogen receptor (ER) agonist, estrogen promotes proliferation in mammary and endometrial cells, maintains bone cell health, and affects lipid metabolism. Estradiol replacement therapy was introduced in the 1960s to counter the decrease in BMD observed in postmenopausal women due to a rapid decline in estradiol levels. However, concerns were raised about the increased risk of estrogen-dependent tumors, such as breast and uterine cancer [26].

Raloxifene is an ER agonist that acts as an agonist on bone tissue, similar to estrogen, while acting as an antagonist on mammary gland cells and endometrial tissue. This selective action minimizes the risk of estrogen-dependent tumors. Hence, it is categorized as an SERM and is recommended for treating osteoporosis in the early postmenopausal period [27]. Both raloxifene and estradiol are fat-soluble drugs capable of crossing the blood–brain barrier via simple diffusion, allowing them to enter the central nervous system. Recent studies have reported several beneficial effects of raloxifene on the central nervous system, such as marked inhibition of brain tissue damage progression in mice with traumatic brain injury [28].

A study investigating the effects of raloxifene on AD indicated a trend toward suppressing cognitive decline when taken for less than a year [29]. However, a large clinical trial conducted in 2015 in patients with mild cognitive impairment (MCI) and AD demonstrated that raloxifene was not effective in preventing the onset of AD [18]. Nevertheless, recent studies have shown that Aβ aggregation already accumulates in the brain tissues of patients with AD in the stage when cognitive decline becomes apparent [19]. This highlights the importance of preclinical therapeutic intervention before Aβ accumulation in brain tissue. Therefore, no clear evidence has been presented regarding the preventive effect of raloxifene on the onset of AD.

Recent research has revealed expression of the estrogen receptor, GPER (a member of the seven transmembrane receptor superfamily), in various cells and tissues, including cell membranes and intracellular organelles such as mitochondria and the endoplasmic reticulum, in addition to the classical ER nuclear receptor. Previous studies have shown that GPER stimulation with estradiol is involved in promoting cell proliferation and differentiation via the activation of the Notch signaling pathway and the Motogen-activated protein kinase (MAPK) pathway [30,31]. Studies on mouse cardiomyocytes have reported that treatment with G-1, a GPER stimulator, suppresses the increase in [Ca^2+^]_i_ induced by L-type calcium channel stimulation and exhibits cardioprotective effects [32]. However, the involvement of GPER and the impact of raloxifene on changes in [Ca^2+^]_i_ in the central nervous system remain unknown.

In our ThT assay experiments, raloxifene demonstrated concentration-dependent inhibition of Aβ_1–42_ aggregation, while estradiol did not inhibit Aβ_1–42_ aggregation (Figure 3). Aβ_1–42_ undergoes folding at the Glu22-Asp23 site, forming a hairpin structure that alters intramolecular polarity and promotes oligomer formation [33]. Previous reports have already indicated the inhibitory effect of raloxifene on Aβ_1–42_ aggregation, suggesting its direct interaction with the N-terminal and intermediate domains of the Aβ_1–42_ peptide, leading to the destabilization of preformed Aβ_1–42_ fibrils [15]. Conversely, estradiol did not exhibit any impact on Aβ_1–42_ aggregation, implying that it does not bind to or act on Aβ_1–42_ peptides as observed with raloxifene. This disparity is likely attributed to the presence of an independent phenolic hydroxyl group in raloxifene, which is believed to inhibit Aβ aggregation [34]. As noted, intracellularly produced APP is cleaved by Aβ monomers via two-step enzymatic reactions and secreted extracellularly. Therefore, it presumably suppresses the extracellular aggregation reaction to Aβo without the effect of estrogen receptors, which are present on the cell membrane or in intracellular organelles.

In clinical practice, the pharmacokinetic concentration of raloxifene is 1.635 ng/mL or 3.45 nM for a single 120 mg dose. The concentration of raloxifene used in this experiment (0.5, 1.0, and 5.0 μM) is 100–1000 times higher than that, which poses a consideration for future studies. However, it is worth noting that even at one-fifth of the concentration of Aβ, raloxifene inhibited aggregation when the concentration of Aβ was 25 μM (Figure 3A). The average Aβ_1–42_ concentration in cerebrospinal fluid is 792 pg/mL or 0.175 nM [35]. Comparatively, one-fifth of this raloxifene concentration amounts to 0.035 nM. Considering that the concentration in the brain is approximately 1/100 of that in the blood and that the total protein concentration in the cerebrospinal fluid is 1/200 of that in blood, the dose required for raloxifene to bind to Aβ and inhibit aggregation in the brain is not noticeably distant from the dose used in clinical practice. However, the accumulation of Aβ in the brain is presumably involved in the pathogenesis of patients with AD, and the concentration of Aβ (Aβ1–40 + Aβ1–42) in the brain of cognitively normal older adults is 130–600 nM [36], which is higher than that in the cerebrospinal fluid, depending on the brain region. The amount of orally ingested raloxifene entering the brain parenchyma has still not been established. Therefore, in the future, it will be necessary to estimate the appropriate dosage by evaluating the in vitro ability of orally ingested raloxifene to enter brain tissue.

In our experiment, both raloxifene and estradiol inhibited the decrease in cell viability induced by Aβo exposure (Figure 4B) and demonstrated inhibition of neuronal cytotoxicity (Figure 5A,H,I). Nuclear ER is expressed in neurons like other cells. Raloxifene, like estradiol, acts as an agonist in neurons [37] and triggers the production of growth factors such as BDNF (brain-derived neurotrophic factor), leading to increased cell viability. However, when conducting similar experiments with pretreatment using fulvestrant, a selective ER blocker, no effect on cell viability was observed compared with non-treating with fulvestrant (Figure 4D). Conversely, pretreatment with G-15, a GPER blocker, prevented the increased viability of Aβo-exposed cells seen with raloxifene and estradiol treatment (Figure 4C). These results suggest that raloxifene and estradiol exert their protective effects against short-term neuronal injury caused by Aβo exposure via a mechanism mediated by GPER stimulation.

Aβo, formed via extracellular aggregation, is believed to induce toxicity by directly damaging neuronal cell membranes [38]. One of the pathological mechanisms involves Aβo directly impairing neuronal cell membranes, forming ion-channel-like pores that alter the membrane potential and result in sustained Ca^2+^ influx into cells [39]. Our study also showed persistently high [Ca^2+^]_i_ levels due to Aβo exposure, but this response was suppressed by pretreatment with Nicardipine, an L-type Ca channel blocker, or MK801, a non-competitive NMDA receptor antagonist (Figure 9A). MK801, like memantine, inhibits the NMDA receptor and exhibits a cell-protective effect [40]. In this experiment, MK801 selectively inhibited the ion-channel portion of NMDA receptors that opened upon Aβo exposure. In the presence of raloxifene and estradiol pretreatment, there was a suppressive effect on the Aβo-induced increase in [Ca^2+^]_i_. These findings suggest that raloxifene or estradiol binds to the nicardipine-binding sites in the Ca^2+^ channel and NMDA receptor, inhibiting the increase in [Ca^2+^]_i_ caused by Aβo. Previous studies using cultured rat cortical neurons have demonstrated that raloxifene inhibits the glutamate-induced increase in [Ca^2+^]_i_ by blocking voltage-gated calcium channels [41].

Inferred from the viability results (Figure 4C,D) of this experiment, the suppressive effect of pretreatment with raloxifene or estradiol on the Aβo-induced increase in intracellular calcium concentration ([Ca^2+^]_i_) could be mediated by GPER on the neural membrane.

However, when G-15 pretreatment was administered before raloxifene or estradiol treatment, it further inhibited the Aβo-induced increase in [Ca^2+^]_i_ compared with the absence of G-15 treatment (Figure 9C,D). GPER stimulation activates phospholipase C, leading to the formation of phosphatidylinositol 4,5 bisphosphate (PIP2), which constitutes cell-membrane phospholipids. As a result, inositol 1,4,5 trisphosphate (IP3) and diacylglycerol (DAG) are formed. These reactions occur at an extremely rapid rate, taking only a few milliseconds, and IP3 facilitates the release of Ca^2+^ from the endoplasmic reticulum. As shown in Figure 9B, treatment with raloxifene or estradiol alone resulted in a transient increase in [Ca^2+^]_i_ 10 s after addition. As previously described, raloxifene or estradiol binds to the nicardipine binding site and the NMDA receptor action site, thereby suppressing the Aβo-induced increase in [Ca^2+^]_i_. However, raloxifene or estradiol transiently increased [Ca^2+^]_i_, and this transient increase was blocked by G-15 pretreatment. In summary, G-15 pretreatment in combination with raloxifene suppressed Aβo-induced increases in [Ca^2+^]_i_ more effectively than raloxifene pretreatment alone.

Oxidative stress induces biochemical changes in neurons and is implicated in the disease progression of many neurological disorders, particularly neurodegenerative diseases such as AD [20]. This experiment also observed various oxidative stress responses, including increased production of ROS due to Aβo exposure, elevated mitochondrial ROS production, and enhanced peroxidation capacity of cell-membrane phospholipids. Treatment with raloxifene and estradiol demonstrated suppression of the oxidative stress response induced by Aβo exposure, including ROS production and mitochondrial ROS production. However, G-15 pretreatment counteracted the inhibition of these oxidative stress responses (Figure 6 and Figure 7). Persistently high levels of [Ca^2+^]_i_ have been shown to overload mitochondria with calcium and increase ROS production [42]. However, considering the effects of GPER treatment on Aβo-induced changes in [Ca^2+^]_i_, it is unlikely that the inhibitory effects of raloxifene and estradiol on Aβo-induced ROS production can be solely explained by mechanisms mediated by membrane GPER stimulation. It has been reported that raloxifene or estradiol increase the expression of Glutathione-SH, an antioxidant in cells [43,44]. Based on these previous studies, the increased expression of antioxidants, including Glutathione-SH, by raloxifene or estradiol may be caused by the stimulation of GPER, which is expressed in intracellular organelles such as mitochondria and the endoplasmic reticulum. In this study, mitochondrial ROS levels increased after 24 h of Aβo exposure in cells treated with G-15 alone compared with those without G-15 treatment. It was speculated that G-15, which had penetrated the cells sufficiently after prolonged exposure, inhibited the basic GPER activity of intracellular organelles, thereby preventing optimum antioxidant function. However, the GPER activity of intracellular organelles is not clear and needs further research.

Aβo exposure also increased membrane phospholipid peroxidation, a type of oxidative stress response. In neuronal cell membranes damaged by Aβo exposure, free radicals initiate a chain reaction with the phospholipid portion of the cell membrane, generating lipid peroxide. This, in turn, leads to structural and functional changes in membrane component proteins, resulting in neuronal cytotoxicity [45]. In this study, when raloxifene or estradiol were administered simultaneously with Aβo, there was a significant inhibition of Aβo-induced phospholipid peroxidation in the raloxifene treatment, which differed significantly from the estradiol treatment. Furthermore, there was no significant difference in the inhibition of phospholipid peroxidation between G-15 pretreatment and no treatment. These findings suggest that raloxifene suppresses Aβo-induced plasma-membrane phospholipid peroxidation without GPER, thereby exerting its neuroprotective effect.

In addition to the previously mentioned ion-channel-like pore formation in neuronal cell membranes, it has been reported that Aβo binds to and disrupts phospholipids in the plasma membrane with its surfactant-like action, owing to its amphiphilic affinity. The N-termini of Aβo tend to interact with the hydrophilic phospholipid heads, while their C-termini penetrate hydrophobic sites [46]. Raloxifene, in addition to its inhibitory effect on Aβ aggregation, has also been reported to destabilize the formed oligomers [15]. This suggests that raloxifene binds to the Aβ oligomer, attenuating its harmful effects on cells, or partially degrades the Aβ-oligomer structure, thereby suppressing the destruction of plasma-membrane phospholipids.

In this experiment, raloxifene inhibited Aβo-induced increases in [Ca2+]i to an extent similar to that exerted by estradiol. Moreover, raloxifene exhibited superior potency to estradiol in inhibiting Aβ aggregation and Aβ-oligomer-induced oxidative stress and exhibited neuroprotective effects against Aβo exposure (Figure 10). The mechanism by which raloxifene treatment significantly improved Aβo exposure-induced cell viability compared with estradiol treatment is attributed to its ability to inhibit Aβo aggregation and promote Aβo degradation [15].

## 5. Conclusions

Raloxifene demonstrated its ability to provide protection against Aβo-induced neuronal cell injury via its inhibitory effects on Aβ aggregation and GPER-mediated antioxidant mechanisms. These findings suggest that raloxifene could potentially play a beneficial role in preventing the onset and progression of AD.

## Figures and Tables

**Figure 1 biomedicines-11-02135-f001:**
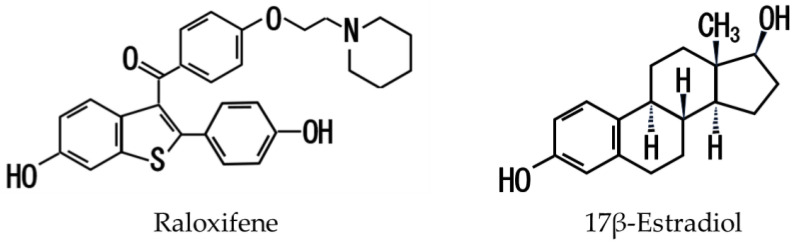
Structures of raloxifene and 17β-estradiol.

**Figure 2 biomedicines-11-02135-f002:**
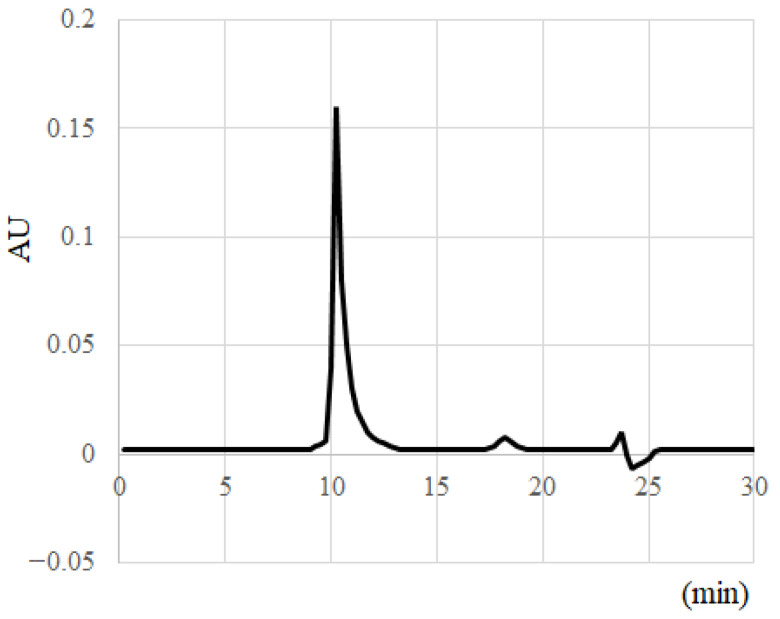
Separation and collection of Aβ molecular species with HPLC.

**Figure 3 biomedicines-11-02135-f003:**
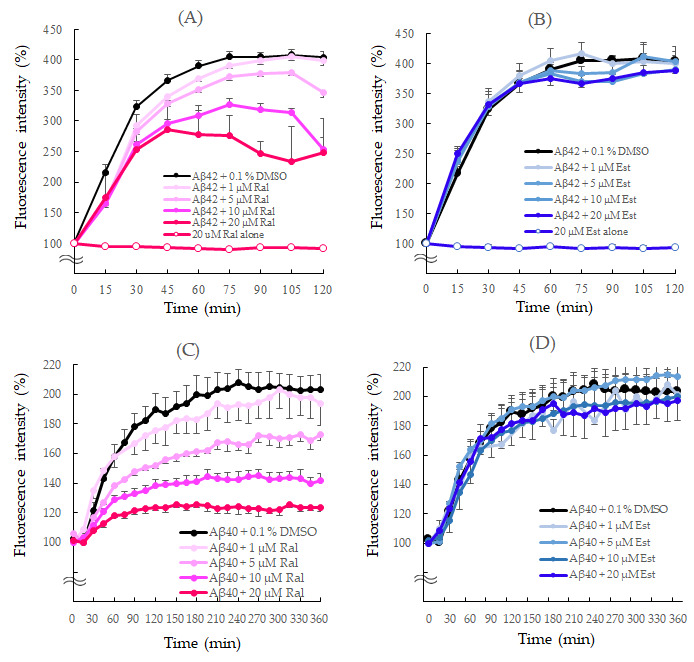
Effects of raloxifene and estradiol on Aβ_1–42_ (**A**,**B**) and Aβ_1–40_ (**C**,**D**) aggregation kinetics. Aggregation kinetics of Aβ_1–40_ and Aβ_1–42_ peptides measured using thioflavin T fluorescence and shown as 100% of the fluorescence intensity at the start of the measurement. (**A**) Aβ_1−42_ (25 μM) with 0.1% DMSO or raloxifene (1, 5, 10, and 20 μM). (**B**) Aβ_1−42_ (25 μM) with 0.1% DMSO or estradiol (1, 5, 10, and 20 μM). (**C**) Aβ_1−40_ (25 μM) with 0.1% DMSO or raloxifene (1, 5, 10, and 20 μM). (**D**) Aβ_1−40_ (25 μM) with 0.1% DMSO or estradiol (1, 5, 10, and 20 μM). The *p*-values in ANOVA were <0.001. Each value represents the mean ± standard error of the mean (SEM) (*n* = 6).

**Figure 4 biomedicines-11-02135-f004:**
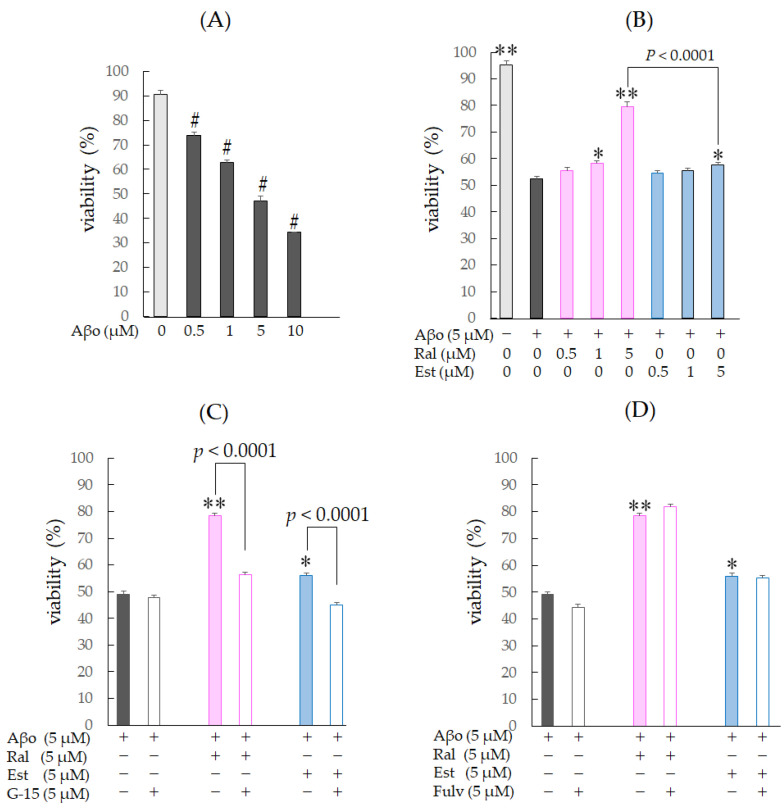
Changes in cell viability in response to Aβo-stimulated SH-SY5Y cells. Viability of Aβo-stimulated SH-SY5Y cells was assessed using the MTT assay. (**A**) SH-SY5Y cells were exposed to Aβo (0.5, 1, 5, and 10 μM) for 3 h. #: *p* < 0.0001 for control versus Aβo-exposed cells (*n* = 8, Dunnett’s). (**B**) SH-SY5Y cells were exposed to Aβo (5 μM) and treated with Aβo + raloxifene (0.5, 1, and 5 μM) or Aβo + estradiol (0.5, 1, and 5 μM) for 3 h. (**C**,**D**) SH-SY5Y cells were pretreated with GPER antagonist (G-15) and nuclear estrogen receptor antagonist (fulvestrant) and treated with Aβo + raloxifene or Aβo + estradiol. +: inclusion of 5 μM Aβo, raloxifene, estradiol; −: non-inclusion. The *p*-values in ANOVA were <0.001. Results are expressed as means + SEMs of 10 individually treated samples. * *p* < 0.05; ** *p* < 0.0001 vs. 5 μM Aβo-exposed cells (*n* = 10, Tukey’s).

**Figure 5 biomedicines-11-02135-f005:**
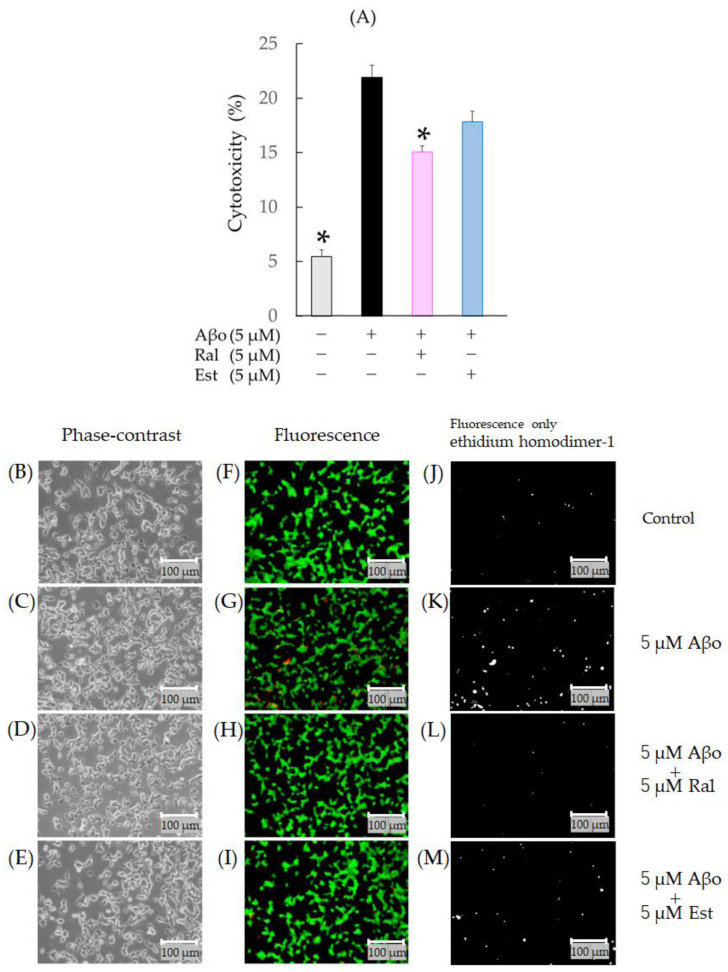
Effect of raloxifene and estradiol on cytotoxicity in Aβo-stimulated SH-SY5Y cells. Cytotoxicity of Aβo-stimulated SH-SY5Y cells was stained with ethidium homodimer-1. (**A**) SH-SY5Y cells were exposed to Aβo (5 μM) and treated with Aβo + raloxifene or Aβo + estradiol (5 μM) for 3 h. In the absence of 5 μM Aβo, the cytotoxicity values of control, 5 μM raloxifene-treated, 5 μM estradiol-treated cells were 5.46 ± 0.6, 5.57± 0.79, and 7.15 ± 0.91%, respectively (no significant differences, *n* = 10, Tukey’s). The *p*-values in ANOVA were <0.001. Measurements were expressed as the means of 10 individually treated samples + SEM. *: *p* < 0.0005 vs. Aβo. (**B**–**I**) Cytotoxicity of Aβo-stimulated SH-SY5Y cells stained with calcein-AM/ethidium homodimer-1 was observed using phase-contrast microscopy (**B**–**E**) and fluorescence microscopy (**F**–**I**). Cytotoxicity of Aβo-stimulated SH-SY5Y cells stained only with ethidium homodimer-1 is shown as gray-scale images (**J**–**M**). Scale bars represent 100 μm.

**Figure 6 biomedicines-11-02135-f006:**
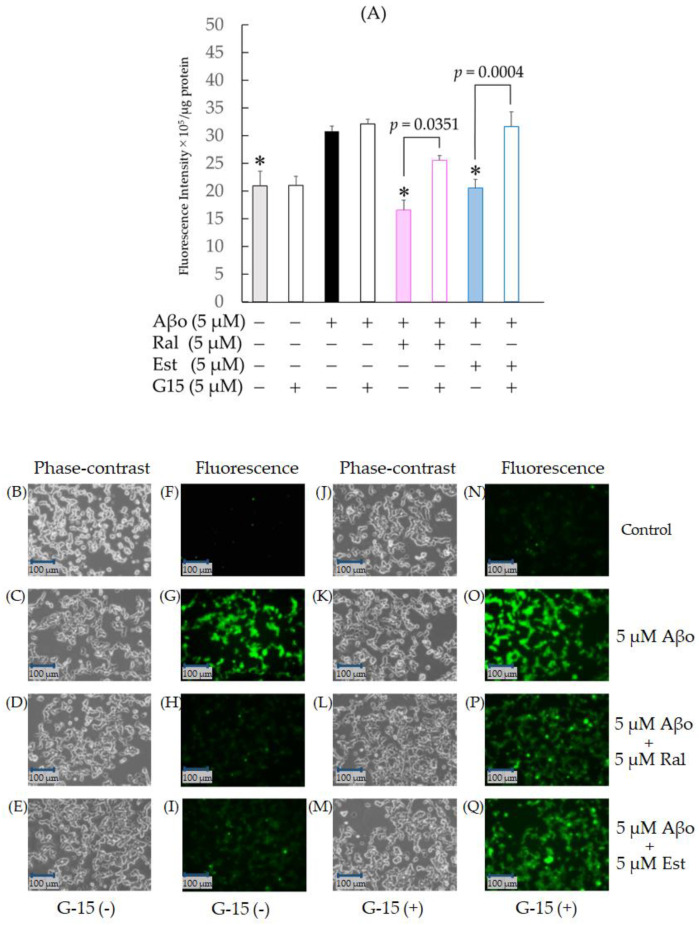
Effect of raloxifene and estradiol on ROS production in Aβo-stimulated SH-SY5Y cells. (**A**) ROS production in SH-SY5Y cells exposed to Aβo was evaluated with CM-H2DCFDA. ROS production in SH-SY5Y cells treated with 5 μM Aβo and 5 μM raloxifene or 5 μM estradiol for 3 h after 5 μM G-15 pretreatment for 30 min was examined. The *p*-values in ANOVA were <0.0001. Measurements were expressed as the means + SEMs of 10 individually treated samples. In the absence of 5 μM Aβo, ROS levels of 5 μM raloxifene-treated and 5 μM estradiol-treated cells were 17.79 ± 1.89 and 19.04 ± 1.36 fluorescence intensity × 10^5^/μg protein, respectively (no significant differences, *n* = 10, Tukey’s). *: *p* < 0.01 vs. 5 μM Aβo. (**B**–**I**) ROS production was observed using phase-contrast microscopy (**B**–**E**) and fluorescence microscopy (**F**–**I**). (**J**–**Q**) ROS production under G-15 pretreatment was observed using phase-contrast microscopy (**J**–**M**) and fluorescence microscopy (**N**–**Q**). Scale bars represent 100 μm.

**Figure 7 biomedicines-11-02135-f007:**
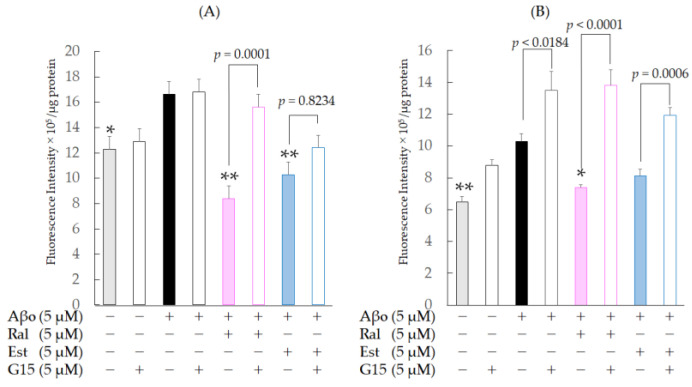
Effect of raloxifene and estradiol on mitochondrial ROS production in SH-SY5Y cells exposed to Aβo. (**A**) SH-SY5Y cells treated with 5 µM Aβo and 5 µM raloxifene or 5 µM estradiol for 3 h were examined. Moreover, cells were pretreated with 5 µM G-15 for 30 min and treated with Aβo + raloxifene or Aβo + estradiol for 3 h. (**B**) SH-SY5Y cells treated with 5 µM Aβo and 5 µM raloxifene or 5 µM estradiol for 24 h were examined. Moreover, cells were pretreated with 5 µM G-15 for 30 min and treated with Aβo + raloxifene or Aβo + estradiol for 24 h. The *p*-values in ANOVA were <0.0001. Measurements were expressed as means + SEMs of 8 individually treated samples. In the absence of 5 μM Aβo, mitochondrial ROS levels of 5 μM raloxifene-treated and 5 μM estradiol-treated cells were 7.67 ± 2.85 and 8.93 ± 4.12 fluorescence intensity × 10^5^/μg protein, respectively (no significant differences, *n* = 10, Tukey’s). * *p* < 0.05, ** *p* < 0.001 vs. 5 μM Aβo.

**Figure 8 biomedicines-11-02135-f008:**
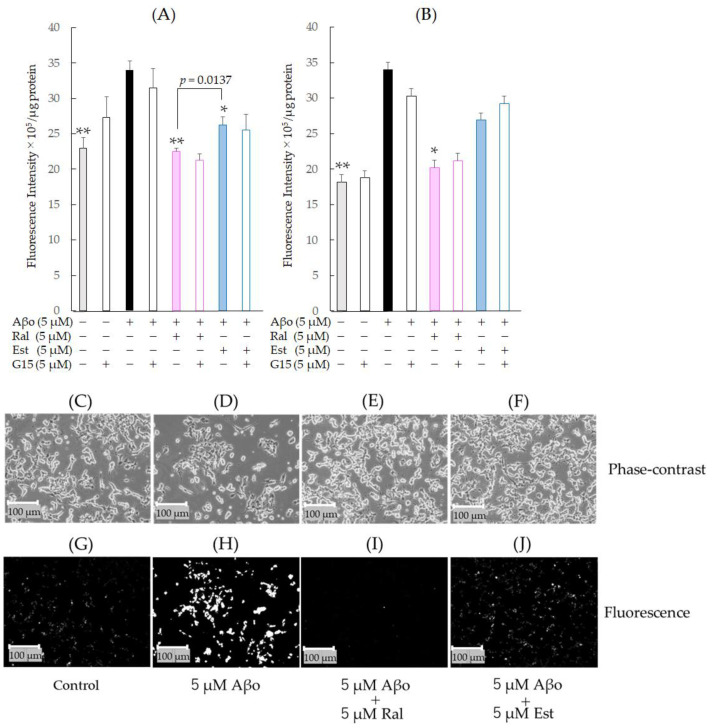
Effect of raloxifene and estradiol on membrane phospholipid peroxidation levels in Aβo-exposed SH-SY5Y cells. (**A**) Membrane phospholipid peroxidation level in Aβo-stimulated SH-SY5Y cells was evaluated with DPPP. SH-SY5Y cells exposed to Aβo (5 μM) and treated with Aβo + 5 μM raloxifene or Aβo + 5 μM estradiol for 30 min were examined. (**B**) Membrane phospholipid peroxidation levels in Aβo-stimulated SH-SY5Y cells were evaluated using DPPP. SH-SY5Y cells exposed to Aβo (5 μM) and treated with Aβo + 5 μM raloxifene or Aβo + 5 μM estradiol for 3 h were examined. Measurements were expressed as the means + SEMs of 8 individually treated samples. In the absence of 5 μM Aβo, membrane phospholipid peroxidation levels of 5 μM raloxifene-treated and 5 μM estradiol-treated cells were 18.82 ± 0.86 and 19.00 ± 1.64 fluorescence intensity × 10^5^/μg protein, respectively (no significant differences, *n* = 8, Tukey’s). The *p*-values in ANOVA were <0.0001. * *p* < 0.05, ** *p* < 0.001 vs. 5 μM Aβo (*n* = 8, Tukey’s). (**C**–**J**) membrane phospholipid peroxidation was also observed using phase-contrast microscopy (**C**–**F**) and fluorescence microscopy (**G**–**J**). Scale bars represent 100 μm. The fluorescence intensity values are shown as gray-scale images.

**Figure 9 biomedicines-11-02135-f009:**
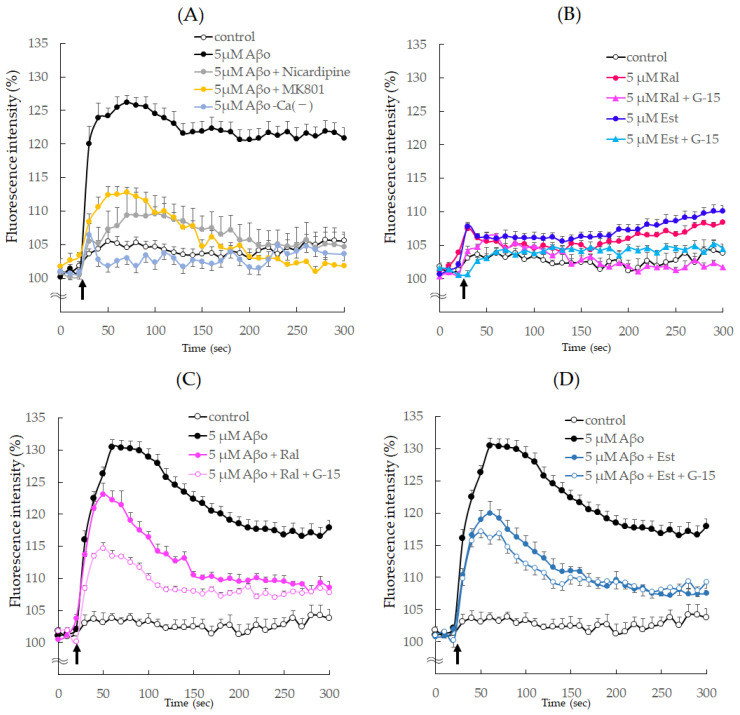
Effect of raloxifene and estradiol on intracellular ionized calcium concentration ([Ca^2+^]_i_) in SH−SY5Y cells. (**A**) The change in [Ca^2+^]_i_ in SH-SY5Y cells after the addition of Aβo was measured using fluorescence intensity for 300 sec. SH-SY5Y cells were exposed to 5 μM Aβo in Ca^2+^-containing or Ca^2+^-less buffer. Furthermore, SH-SY5Y cells were exposed to 5 μM Aβo after calcium-channel antagonist (10 μM nicardipine) or an NMDA receptor blocker (10 μM MK801) pretreatment for 10 min. (**B**) SH-SY5Y cells were supplemented with raloxifene or estradiol after G-15 pretreatment or no pretreatment. (**C**) SH-SY5Y cells treated with raloxifene were exposed to 5 μM Aβo after G-15 pretreatment or no pretreatment. (**D**) SH-SY5Y cells treated with estradiol were exposed to 5 μM Aβo after G-15 pretreatment or no pretreatment. The time of administration is indicated by an arrow. Fluorescence intensity was expressed as 100% of the value at the onset.

**Figure 10 biomedicines-11-02135-f010:**
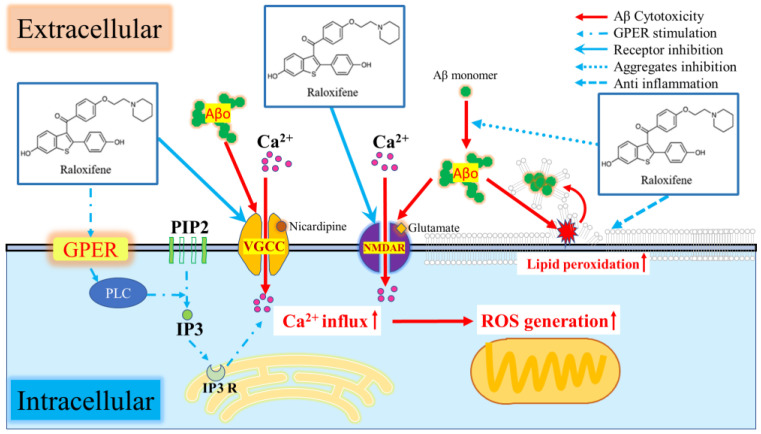
Schematic of the neurotoxic mechanism of Aβo and inhibition by raloxifene.

## Data Availability

Not applicable.

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
