# Peer review of "Neuroprotective Potential of Raloxifene via G-Protein-Coupled Estrogen Receptors in Aβ-Oligomer-Induced Neuronal Injury"

_biomedicines, 2023, doi:10.3390/biomedicines11082135_

Round 1
Reviewer 1 Report
This manuscript describes a series of experiments that investigate whether the SERM Raloxifene can protect cells against neurotoxic forms of beta amyloid oligomers (Abos) linked with Alzheimer’s Disease (AD). As noted by the authors, Studies have suggested that Raloxifene has neuroprotective effects in a variety of models of brain injury and disease, although an important clinical trial found that Raloxifene did not improve cognitive function in AD patients. Nevertheless, the possibility that Raloxifene might be beneficial when given prior to the onset of AD symptoms (in a ‘protection’ protocol) deserves to be investigated. In the current study, the authors use SH-SY5Y neuroblastoma cells as a convenient surrogate for real neurons, based on past studies showing that this line can model the deleterious effects of Abos on different aspects of cell viability. Using established assays and well-controlled methods, they have made two important discoveries: (1) that Raloxifene inhibited the aggregation of synthetic Ab into higher molecular weight oligomers at least in vitro; and (2) that treatment with Raloxifene protected SH-SY5Y cells against the deleterious effects of exogenous Abos, including beneficial effects on cell viability, ROS production, lipid peroxidation, and calcium overload.
The manuscript is well written (only two minor text suggestions were noted); the figures are clear and (for the most part) described appropriately in the text and legends; the statistical methods are valid; and the interpretation of results are reasonable. Overall, this work provides an interesting new perspective on how SERMs like Raloxifene might protect against the deleterious effects of neurotoxic proteins linked with AD, including beta amyloid aggregates. They also support the hypothesis that administering Raloxifene prior to the onset of cognitive clinical symptoms might be beneficial for patients at risk of AD.
No major issues were identified in the manuscript and figures. However, the following points should be considered by the authors in their revisions.
1. The observation that Raloxifene can inhibit Ab aggregation in vitro (in the absence of cells or tissues) is intriguing, but it also raises questions about how this effect might contribute to the neuroprotective effects of Raloxifene in vivo. Given the ongoing controversy about where Ab oligomerization is initiated (intracellularly or extracellularly), the authors might add a short discussion about the relative importance of this potential action of Raloxifene (within brain parenchyma) versus its better characterized effects as a SERM.
2. In figure 2: since estradiol also activates GPER (as well as conventional ERs), it is surprising that estradiol treatment did not significantly improve viability in Abo-treated cells (panel 4B), although co-treatment with estradiol and G-15 significantly reduced viability compared to estradiol alone (panel 4C). By comparison, estradiol treatment was clearly protective in their ROS assays (figure 6). It would be appropriate for the authors to comment on this curious paradox.
3. In figure 7: It is somewhat surprising that G-15 alone causes a significant increase in mitochondrial ROS. The authors might add a brief comment about how they interpret this result. Does it suggest that basal GPER activity normally protects against the generation of mitochondrial ROS in otherwise untreated cells?
4. In the discussion (related to the schematic in Figure 10): The authors provide an appropriate and informative discussion of how their results can be interpreted, based on the model that GPER is predominantly localized to the plasma membrane. However, as they undoubtedly know, there is strong evidence that GPER can also localize to a variety of intracellular compartments, including mitochondrial membranes. I do NOT suggest that they add this feature to the model in Figure 10 (that would be too complicated), but it would be helpful for them to add a brief discussion about the potential contribution of intracellular GPER activation in the context of their different assays (e.g. ROS production versus membrane lipid peroxidation).
Suggestions for Figures:
· Figure 4: Please check the x-axis labeling in panel 4D: I believe that the gray histogram should be labeled with a minus sign (‘-‘), not a plus sign (‘+’) since presumably the cells were not treated with Abo (compared to the black histogram).
· Figure 4D, 5A, 6A, 7, 8A : Please add comparison bars (with appropriate asterisks) to indicate which histograms are significantly different. Note that this convention is used for some comparisons but not others. Adding the missing comparison bars will help make the figures more clear, and will relate well to the descriptions in the text.
· Figure 5: I suggest that the authors add an additional set of panels next to F-I, showing only the red channel signal as gray scale images. Currently it is somewhat difficult to see the red channel signals in the predominantly green images, and this result deserves to be illustrated more clearly.
· Figure 8F-I: Please consider changing these panels from blue to gray scale images; this approach will make it easier to visualize the differences in cell labeling.
Minor text issues:
1. On p. 7, para. 2, last sentence (ending in ‘Figure 4B’). Please doublecheck this sentence, it appears to be incomplete (what does 5 uM Estradiol do?)
2. On page 15, last paragraph, sentence beginning with ‘Although’: I suggest either remove the word ‘Although’ (it is not necessary) or incorporate this phrase into the adjacent sentences, as currently it is grammatically incomplete.
Author Response
Manuscript ID: biomedicines-2507333
Reply to Reviewers
Dear Reviewer #1,
Thank you very much for your mail dated 15 July 2023 regarding the reviewers’ comments on our manuscript titled “The Neuroprotective Potential of Raloxifene through G Protein-Coupled Estrogen Receptors in Aβ Oligomer-Induced Neuronal Injury” (Manuscript ID: biomedicines-2507333). Please accept our sincere gratitude to you and the two respectable reviewers for the encouraging feedback. A point by point response has been provided for each query. The corresponding changes in the main manuscript have been marked in red font and changes to figures have been marked by enclosing the revisions in red squares.
Answers to the Reviewer #1
Thank you very much for your pertinent and productive comments. We have revised the manuscript accordingly. We sincerely apologize for the typographical and English language errors, and we are very thankful for your careful reading.
Major concerns
- The observation that Raloxifene can inhibit Ab aggregation in vitro (in the absence of cells or tissues) is intriguing, but it also raises questions about how this effect might contribute to the neuroprotective effects of Raloxifene in vivo. Given the ongoing controversy about where Ab oligomerization is initiated (intracellularly or extracellularly), the authors might add a short discussion about the relative importance of this potential action of Raloxifene (within brain parenchyma) versus its better characterized effects as a SERM.
Response: We appreciate this insightful comment. As pointed out, we also agree that it is important to determine how the Aβ aggregation inhibitory effect of Raloxifene observed in the study would exhibit neuroprotective effects in vivo. Based on established research, we believe that Aβ is secreted extracellularly after a two-step enzymatic cleavage of intracellular APP. We have added the following statements to the Discussion section, lines 536-539.
“As noted, intracellularly produced APP is cleaved by Aβ monomers through two-step enzymatic reactions and secreted extracellularly. Therefore, it presumably suppresses the extracellular aggregation reaction to Aβo without the effect of estrogen receptors, which present on the cell membrane or in intracellular organelles.”
- In figure 2: since estradiol also activates GPER (as well as conventional ERs), it is surprising that estradiol treatment did not significantly improve viability in Abo-treated cells (panel 4B), although co-treatment with estradiol and G-15 significantly reduced viability compared to estradiol alone (panel 4C). By comparison, estradiol treatment was clearly protective in their ROS assays (figure 6). It would be appropriate for the authors to comment on this curious paradox.
Response: We apologize if the data conveyed such a remark. We express regret for not considering the contradiction between the effect of estradiol on suppressing ROS production and improving cell viability in Aβo-treated cells. We speculated that the various cytotoxic effects of Aβo overpowered the ROS production inhibitory effects of estradiol and did not contribute to improving cell viability. Based on our experimental results, we speculated that raloxifene treatment enhanced cell viability by improving Aβo toxicity by exerting its anti-aggregation effects during the 3 h incubation. I have added the following statements (lines 645–651) in the discussion section regarding difference in neuroprotective action between raloxifene and estradiol treatments.
“In this experiment, raloxifene inhibited Aβo-induced increases in [Ca2+]i to a similar extent as that by estradiol. Moreover, raloxifene exhibited superior potency to estradiol in inhibiting Aβ aggregation and Aβ oligomer-induced oxidative stress and exhibited neuroprotective effects against Aβo exposure (Figure 10). The mechanism by which raloxifene treatment significantly improved Aβo exposure-induced cell viability compared with estradiol treatment is attributed to its ability to inhibit Aβo aggregation and promote Aβo degradation [34]. ”
- In figure 7: It is somewhat surprising that G-15 alone causes a significant increase in mitochondrial ROS. The authors might add a brief comment about how they interpret this result. Does it suggest that basal GPER activity normally protects against the generation of mitochondrial ROS in otherwise untreated cells?
Response: Thank you for the opportunity to provide clarity here. As mentioned, the basal GPER activity of SH-SY5Y cells was believed to suppress the generation of mitochondrial ROS in untreated cells. As noted by reviewer 2, we also newly presented the results of exposing the reagent for 3 h, but interestingly, we did not observe an increase in mitochondrial ROS levels due to G-15 treatment alone for a short time of 3 h. We presumed that the difference was related to the fact that long-term exposure to G-15 broadly inhibited GPER in intracellular organelles in addition to the GPER inhibitory effect on cell membranes. This explanation has been added to the discussion section, lines 618–623 of the discussion.
- In the discussion (related to the schematic in Figure 10): The authors provide an appropriate and informative discussion of how their results can be interpreted, based on the model that GPER is predominantly localized to the plasma membrane. However, as they undoubtedly know, there is strong evidence that GPER can also localize to a variety of intracellular compartments, including mitochondrial membranes. I do NOT suggest that they add this feature to the model in Figure 10 (that would be too complicated), but it would be helpful for them to add a brief discussion about the potential contribution of intracellular GPER activation in the context of their different assays (e.g. ROS production versus membrane lipid peroxidation).
Response: Thank you very much for this valuable suggestion. We agree with your opinion, and have accordingly specified that GPER is expressed not only in the cell membrane but also in intracellular organelles such as the mitochondria and endoplasmic reticulum. The insertions have been done in lines 515–518 and lines 618–623 of the discussion section.
Suggestions for Figures:
Figure 4: Please check the x-axis labeling in panel 4D: I believe that the gray histogram should be labeled with a minus sign (‘-‘), not a plus sign (‘+’) since presumably the cells were not treated with Abo (compared to the black histogram).
Response: We apologize for the oversight. We have replaced the (‘+’) sign with a (‘-‘)sign in Figure 4B in the legend. We will ensure vigilance in future.
Figure 4D, 5A, 6A, 7, 8A : Please add comparison bars (with appropriate asterisks) to indicate which histograms are significantly different. Note that this convention is used for some comparisons but not others. Adding the missing comparison bars will help make the figures more clear, and will relate well to the descriptions in the text.
Response: Thank you for this valuable suggestion. For Figures 4D, 5A, 6A, 7, and 8A, we have added bars (with appropriate asterisks) for indicating significant differences and have expanded the notation for better visibility.
Figure 5: I suggest that the authors add an additional set of panels next to F-I, showing only the red channel signal as gray scale images. Currently it is somewhat difficult to see the red channel signals in the predominantly green images, and this result deserves to be illustrated more clearly.
Response: Thank you for this constructive suggestion. We agree with your comment and have added an additional set of panels (J-M) next to F-I, showing only the red channel signal as gray scale images (Figure J-M).
Figure 8F-I: Please consider changing these panels from blue to gray scale images; this approach will make it easier to visualize the differences in cell labeling.
Response: Thank you for this valuable suggestion. We completely agree with your proposal and have changed the blue fluorescence images to gray scales (Figure8G-J).
Minor text issues:
On p. 7, para. 2, last sentence (ending in ‘Figure 4B’). Please doublecheck this sentence, it appears to be incomplete (what does 5 uM Estradiol do?)
Response: We apologize for this typographical error. We have added the phrase “showed significant increase in cell viability” to this statement.
On page 15, last paragraph, sentence beginning with ‘Although’: I suggest either remove the word ‘Although’ (it is not necessary) or incorporate this phrase into the adjacent sentences, as currently it is grammatically incomplete.
Response: We apologize for the error. As suggested, we have removed the term "Although" from the text.
Reviewer 2 Report
The study proposed by Dr. Nohara and colleagues investigates the effects of raloxifene (Ral) and estradiol in SH-SY5Y cells exposed to amyloid-b oligomers (Abo) and concludes that Ral provides protection against Abo neurotoxicity by inhibiting the aggregation of Ab and, moreover, by activating GPER-mediated antioxidant mechanisms.
Although the investigated topic is of interest, the results presented do not support the conclusions reached.
Specifically (in order of appearance):
1) Regarding Abo isolation by HPLC: what was used as a standard to identify the Abo peak? Figure 2 proves nothing without the proper references and controls.
2) Besides the fact that the inhibitory effect of Ral on Ab aggregation has already been demonstrated (ref. 32 in the manuscript), experiments in Fig 3 should include, as a control, a sample with thioflavin T +/- Ral, without Ab.
3) Results of the MTT assay are generally expressed in terms of percentage of cell viability compared to untreated control cells and calculated as OD sample/OD control x 100. In this study (Fig. 4), strangely, the raw absorbance values are plotted. Again regarding the MTT results, what is the difference between the treatments in the first two bars of Fig. 4B?
4) The microscope images (Figs. 5 and 6) are of low quality and do not allow a clear evaluation of the results obtained.
5) In figures 5A, 6A, and 7 the results are expressed in terms of fluorescence/mg protein. Since the fluorescence was measured on the plated cells, it is conceivable that the proteins were measured retrospectively. This point should be clarified.
6) Cell viability, cytotoxicity, and ROS were measured after 3 hours of treatment, mitochondrial ROS after 24 hours of treatments, and lipid peroxidation after 30 minutes of treatment. Why?
7) As far as the effects of Ral on the intracellular Ca2+ concentration are concerned, the reported results indicate a reduction of about 8% compared to the Ab effect (Fig. 9C), and an increase of about 10% with Ral alone. What value can these small variations have, even considering that the standard deviations are missing?
8) In the discussion (last paragraph on page 14), the authors admit that the Ral concentration used in their study is 100-1000 times higher than the pharmacokinetic concentration of 3.45 nM obtained with a Ral dose in the clinical practice (120 mg). Then, to justify this difference, they propose an empirical calculation based on an average concentration of Ab in the CSF equal to 0.175 nM, which would support the efficacy of 3.45 nM Ral. As far as we know, however, Ab in the brain, rather than cerebrospinal fluid, correlates with neurocognitive impairment in AD. According to Naslung et al. (JAMA, 2000), the concentration of Ab (Ab40 + Ab42) in the cognitively normal elderly human brain ranges from 130 to 600 nM, depending on the brain areas.
Minor editing of English language.
Author Response
Dear Reviewer #2,
Thank you very much for your mail dated 15 July 2023 regarding the reviewers’ comments on our manuscript titled “The Neuroprotective Potential of Raloxifene through G Protein-Coupled Estrogen Receptors in Aβ Oligomer-Induced Neuronal Injury” (Manuscript ID: biomedicines-2507333). Please accept our sincere gratitude to you and the two respectable reviewers for the encouraging feedback. A point by point response has been provided for each query. The corresponding changes in the main manuscript have been marked in red font and changes to figures have been marked by enclosing the revisions in red squares.
Answers to the Reviewer #2
Thank you very much for your valuable and constructive comments. We have revised the manuscript accordingly. We sincerely apologize for the typographical and English language errors, and we are very thankful for your careful reading.
1) Regarding Abo isolation by HPLC: what was used as a standard to identify the Abo peak? Figure 2 proves nothing without the proper references and controls.
Response: We apologize if the data conveyed such a remark. We regret the absence of a control sample for Aβo peak identification using HPLC. Hence, we have referred to a previous analysis (ref [20, 21]). In an experiment using the same protocol, the protein peak at an approximate retention time of 10 minutes was isolated and analyzed using an electron microscope. It was confirmed that the peak corresponded to Aβo.
2) Besides the fact that the inhibitory effect of Ral on Ab aggregation has already been demonstrated (ref. 32 in the manuscript), experiments in Fig 3 should include, as a control, a sample with thioflavin T +/- Ral, without Ab.
Response: Thank you for your valuable suggestion. Accordingly, we have presented modified Figures 3A-B by adding experimental results with samples containing ThT/Ral without Aβ as a control. In experiments without Aβ, the addition of ThT did not increase the fluorescence intensity.
3) Results of the MTT assay are generally expressed in terms of percentage of cell viability compared to untreated control cells and calculated as OD sample/OD control x 100. In this study (Fig. 4), strangely, the raw absorbance values are plotted. Again regarding the MTT results, what is the difference between the treatments in the first two bars of Fig. 4B?
Response: Thank you for the opportunity to provide clarity here. As pointed out by you, it is common to use the absorbance of untreated cells to express the percentage of cell viability when calculating the cell viability using the MTT assay. Accordingly, we have mentioned considering the absorbance of SH-SY5Y cells cultured in DMEM/Ham's F-12 medium containing 10% FBS as 100% viability and that for cells stimulated by exposure to 1% saponin as 0% viable.
We apologize for the typographical error in the first two bars of Figure. 4B. We have replaced the (‘+’) sign with a (‘-‘)sign in the X-axis legend of Figure 4B on the extreme left.
4) The microscope images (Figs. 5 and 6) are of low quality and do not allow a clear evaluation of the results obtained.
Response: We apologize if the data conveyed such a remark. The microscope images of Figure 5 and 6 have been enlarged for better visibility. In addition, as suggested by the reviewers, the ethidium homodimer-1 staining images have been further clarified by presenting them as gray scale images (Figure 5J-M).
5) In figures 5A, 6A, and 7 the results are expressed in terms of fluorescence/mg protein. Since the fluorescence was measured on the plated cells, it is conceivable that the proteins were measured retrospectively. This point should be clarified.
Response: Thank you for the opportunity to provide clarity here. The following text was added to the Materials and Methods sections:
“Following fluorescence measurements, the cellular protein content on the plate was determined using the Bio-Rad Protein Assay Dye Reagent Concentrate (Bio-Rad Laboratories, Inc), and the results were represented as the fluorescence intensity per unit protein.”
6) Cell viability, cytotoxicity, and ROS were measured after 3 hours of treatment, mitochondrial ROS after 24 hours of treatments, and lipid peroxidation after 30 minutes of treatment. Why?
Response: Thank you for the opportunity to provide clarity here. Based on the results of the cell viability experiment, the experiment was conducted with the drug exposure time set to 3 h. DPPP was measured in 30 min because it responds quickly after drug exposure, but we added the drug exposure results for 3 h in response to your comments (Figure 8A-B). The mitochondrial ROS experiment findings for 24-h exposure have been presented since these results showed a stronger significant difference; however, results of the 3 h exposure are also presented (Figure 7A-B).
7) As far as the effects of Ral on the intracellular Ca2+ concentration are concerned, the reported results indicate a reduction of about 8% compared to the Ab effect (Fig. 9C), and an increase of about 10% with Ral alone. What value can these small variations have, even considering that the standard deviations are missing?
Response: Thank you for the pertinent query. Regarding Figure .9, we have added the standard deviation to all results. As pointed out by you, a decrease in intracellular Ca2+ concentration by approximately 8% was observed after pretreatment with Ral, compared to that on exposure to Aβo alone. Although the changes were minimal, Ral pretreatment showed a decrease in intracellular Ca2+ concentration induced by Aβo exposure, with significant differences at all time point from 60 to 300 seconds from the start of measurement.
8) In the discussion (last paragraph on page 14), the authors admit that the Ral concentration used in their study is 100-1000 times higher than the pharmacokinetic concentration of 3.45 nM obtained with a Ral dose in the clinical practice (120 mg). Then, to justify this difference, they propose an empirical calculation based on an average concentration of Ab in the CSF equal to 0.175 nM, which would support the efficacy of 3.45 nM Ral. As far as we know, however, Ab in the brain, rather than cerebrospinal fluid, correlates with neurocognitive impairment in AD. According to Naslung et al. (JAMA, 2000), the concentration of Ab (Ab40 + Ab42) in the cognitively normal elderly human brain ranges from 130 to 600 nM, depending on the brain areas.
Response: We thank the reviewer for the careful reading. As pointed out by you, the accumulation of Aβ in the brain is presumably involved in the pathology of patients with AD. To our knowledge, there are no reports to verify the translocation of raloxifene to brain tissue in vivo, and we consider it necessary to verify this in future. We have added the following information to the discussion, lines 550–556.
“However, the accumulation of Aβ in the brain is presumably involved in the pathogenesis of patients with AD, and the concentration of Aβ (Aβ1–40 + Aβ1–42) in the brain of cognitively normal older adults is 130–600 nM [37], which is higher than that in the cerebrospinal fluid, depending on the brain region. The amount of orally ingested raloxifene actually entering the brain parenchyma, has still not been established. Therefore, in future, it is necessary to estimate the appropriate dosage by evaluating the in vitro ability of orally ingested raloxifene to enter the brain tissue.”
Comments on the Quality of English Language
Minor editing of English language.
Response: We sincerely apologize for the typographical and English language errors, and we are very thankful for your careful reading. Our revised manuscript has been edited by an English language editing service.

Round 2
Reviewer 2 Report
The manuscript has been modified in accordance with my request.